# The Role of MicroRNAs in HER2-Positive Breast Cancer: Where We Are and Future Prospective

**DOI:** 10.3390/cancers14215326

**Published:** 2022-10-29

**Authors:** Valentina Fogazzi, Marcel Kapahnke, Alessandra Cataldo, Ilaria Plantamura, Elda Tagliabue, Serena Di Cosimo, Giulia Cosentino, Marilena V. Iorio

**Affiliations:** Fondazione IRCCS Istituto Nazionale dei Tumori, 20133 Milan, Italy

**Keywords:** breast cancer, microRNA, HER2

## Abstract

**Simple Summary:**

Breast cancer is the most diagnosed malignancy in woman worldwide and, despite the availability of new innovative therapies, it remains the first cause of death for tumor in woman. 20% of all breast cancer cases are HER2 positive, meaning that they are characterized by an aberrant expression of the growth factor receptor HER2. This receptor is involved in survival and proliferation mechanisms, conferring to this breast cancer subtype a particular aggressiveness. The introduction of anti-HER2 agents, such as trastuzumab, in the clinical practice, significantly improved the prognosis. However, a great portion of patients is not responsive to this therapy. Thus, cancer research is working to provide new tools to better manage HER2 positive breast cancers, such as biomarkers and therapeutic approaches. MicroRNAs could be used for these purposes. They are small molecules involved in almost all biological processes, including cancer promoting pathways. Researchers consider microRNAs as promising clinical tools because they are easily detectable and stable in both tissues and blood samples, and an increasing body of evidence supports their potential use as targets of therapy, prognostic and predictive biomarkers, or therapeutic agents. This review sums up the most recent scientific publications about microRNAs in HER2 positive breast cancer.

**Abstract:**

Breast cancer that highly expresses human epidermal growth factor receptor 2 (HER2+) represents one of the major breast cancer subtypes, and was associated with a poor prognosis until the introduction of HER2-targeted therapies such as trastuzumab. Unfortunately, up to 30% of patients with HER2+ localized breast cancer continue to relapse, despite treatment. MicroRNAs (miRNAs) are small (approximately 20 nucleotides long) non-coding regulatory oligonucleotides. They function as post-transcriptional regulators of gene expression, binding complementarily to a target mRNA and leading to the arrest of translation or mRNA degradation. In the last two decades, translational research has focused on these small molecules because of their highly differentiated expression patterns in blood and tumor tissue, as well as their potential biological function. In cancer research, they have become pivotal for the thorough understanding of oncogenic biological processes. They might also provide an efficient approach to early monitoring of tumor progression or response to therapy. Indeed, changes in their expression patterns can represent a flag for deeper biological changes. In this review, we sum up the recent literature regarding miRNAs in HER2+ breast cancer, taking into account their potential as powerful prognostic and predictive biomarkers, as well as therapeutic tools.

## 1. Introduction

### 1.1. HER2-Positive Breast Cancer

Breast cancer is the most common malignancy worldwide, with a lifetime risk of 1 in 8 individuals (approximately 13%), and is responsible for 13.6 deaths in 100,000 [1]. Most breast malignancies arise from epithelial elements and are categorized as carcinomas. Based on immunohistochemistry, breast cancer is classified as hormone receptor positive (HR+), human epidermal growth factor receptor 2 positive (HER2+), and triple-negative (TNBC). Furthermore, the proliferation marker Ki-67 and histological grading allow estimation of the tumor aggressiveness [2]. Because of the highly heterogeneous molecular alterations of the disease, Perou et al. defined a classification based on gene expression patterns, dividing breast cancer into five molecular subtypes: luminal A, luminal B, HER2-enriched, basal-like, and normal-like, each characterized by a different clinical behavior [3,4]. In particular, the two luminal subtypes are HR+, differing from each other in the expression of HER2 and the Ki-67 score; indeed, luminal A tumors (50–60% of all breast cancer cases) are HER2- with a Ki-67 score < 20%, whilst most luminal B tumors are HER2+ with a Ki-67 score ≥ 20%. The basal-like subtype represents 15% of all breast cancers and most of them (75%) are TNBC, while normal-like malignances (5–10% of cases) are characterized by the expression of adipose tissue-related genes. As the focus of this review, the features of the HER2-enriched subtype will be widely described in the following paragraphs [3,4].

HER2, encoded by the gene *ERBB2* (erythroblastic oncogene B), is a tyrosine kinase receptor. It is overexpressed in approximately 15–20% of all breast cancer cases [5] and associated with poor prognosis, especially before the introduction of targeted therapies [6]. The first available HER2-targeted drug was the humanized monoclonal antibody trastuzumab [7]. The remarkable clinical success of this drug led to the development and approval of additional HER2-specific drugs such as other monoclonal antibodies including pertuzumab and margetuximab, antibody-drug conjugates as trastuzumab-emtansine (T-DM1) and trastuzumab-deruxtecan (T-DXd), or receptor tyrosine kinase inhibitors such as lapatinib, neratinib, and tucatinib. Initially, most of these drugs were approved only for the metastatic disease combined with chemotherapy. However, further clinical trials revealed an outstanding impact in early disease stages as well. Up-to-date patients with early non-metastatic HER2+ breast cancer receive neoadjuvant chemotherapy combined with HER2-targeted therapy. Neoadjuvant treatment is followed by tumor removing surgery and, in the case of breast conserving therapy, subsequent radiation therapy [8]. Depending on whether or not a pathological complete response (pCR) is achieved after primary neoadjuvant treatment, either the same anti-HER2 therapy will be continued until completion of 1 year or, in case of non-pCR, the HER2-targeted regimen will be modified.

Even though the application of these drugs improved pCR rates and event-free survival (EFS), there are still approximately 50% of HER2+ breast cancer patients that do not benefit from this therapy [9]. Due to the potential toxicity of most HER2-targeted drugs (cardiotoxicity, skin rash, pain, insomnia), a compelling need of clinical research is to identify responding patients early using reliable predictive biomarkers in order to avoid overtreatment, and encourage patients with promising predictive values to continue in spite of temporarily debilitating side effects.

### 1.2. Trastuzumab Resistance

Since its approval for the first-line treatment of metastatic disease in the late nineties, trastuzumab still represents the cornerstone for the treatment of patients with HER2+ breast cancer. It is a recombinant humanized monoclonal antibody targeting the extracellular domain 4 of HER2 [10]. The anti-tumor effect is based on many different mechanisms of action including blocking of the HER2 receptor, inhibition of neoangiogenesis, receptor internalization and degradation, induction of antibody-dependent cell-mediated cytotoxicity (ADCC), and inhibition of DNA repair [11,12]. Unfortunately, both primary and secondary resistance to trastuzumab exist, as reported in about one-third of women destined to never respond to treatment [9,13], and in those whose cancer inevitably progresses about a year and a half after starting treatment [14].

Resistance to trastuzumab can be developed through many different processes, implying the existence of more than a single mechanism of action. The efficacy of trastuzumab can be impaired by epitope masking (via MUC4 or CD44/polymeric hyaluronan complex), thus disabling the binding of trastuzumab to the extracellular domain. A truncated form of HER2, p95HER2, lacking the extracellular binding domain for trastuzumab but maintaining intracellular kinase activity, has also been described. The activation of HER2 is followed by a downstream signaling cascade. In trastuzumab-resistant cells, these downstream pathways can be found upregulated, even though there is no appropriate stimulus by the HER2 receptor. The decreased activation of HER2 by trastuzumab can also be compensated for by the activation of other oncogenic receptors, such as the β2-adrenic receptor, IGF1-receptor, or other members of the epidermal growth receptor family. Furthermore, many different mechanisms have been described leading to an impairment of ADCC in trastuzumab resistant cells, due to a lower recognition of the Fc fragment of trastuzumab bound to its epitope [10].

### 1.3. MicroRNAs

MicroRNAs (miRNAs) are small oligonucleotides of approximately 20 nucleotides. These molecules are known to play an important role in the post-transcriptional regulation of gene expression, inhibiting the translation of the target messenger-RNAs (mRNAs) [15].

First, miRNAs were thought to be synthesized exclusively by a process currently known as the canonical pathway, whereas other non-canonical processes, such as the mirtron pathway, were discovered later [16]. Canonical biogenesis consists of the two-step cleavage of the long primary miRNA transcript (pri-miRNA) by means of the RNase II enzyme DROSHA into pre-miRNAs containing the characteristic hairpin structure. After transportation into the cytosol, the second cleavage step is performed; the multi-domain enzyme DICER cleaves the double-strand pre-miRNA stem in an asymmetrical manner into the mature duplex miRNAs.

After the association with argonaute proteins (AGO1, AGO2, AGO3, or AGO4) and other proteins to form the miRNA-induced silencing complex (miRISC), the double-strand miRNA is hydrolyzed into the guide strand recognizing the target mRNA and the passenger strand, which is released into the cytosol. Originally, the passenger strand was thought to be useless waste ready for degradation, but it was later demonstrated to be able to act as a miRNA in the same way as the guide strand [17]. To differentiate two miRNAs originating from the same pre-miRNA, the non-predominant form will receive a *. If the data is not sufficient to define which strand is predominant, the strand originating from the 5′ arm receives the suffix -5p and the strand originating from the 3′ arm receives the suffix -3p.

Once the miRISC complex is assembled, the mature miRNA guides the interaction with the target mRNA. More specifically, the miRNA “seed” region, defined as nucleotides 2–8 at the 5′ end, is able to recognize complementary sequences that can be located at the 3′-UTR, 5’UTR, and/or the open reading frame (ORF) of the target mRNA. When the binding takes place, the subsequent translation can be inhibited or the whole mRNA can be degraded. There are many prediction tools (e.g., TargetScan, Tools4miRs) to identify putative targets of one miRNA, but the gold standard to confirm the targeting in vitro remains the luciferase reporter assay.

Frequently, miRNA genes are located at fragile sites in the genome [18]. Several miRNA genes are clustered and co-expressed as polycistronic units, suggesting a functional connection. Furthermore, most miRNAs reside in introns of their host genes; thus, they may be expressed together with the adjacent protein-coding sequences. Nevertheless, a number of miRNA-coding genes are dispersed in the genome with an independent transcription and their own promoters.

Currently, it is well known that miRNAs play different and crucial roles in both biological and pathological processes, including cancer. The first evidence of miRNA involvement in cancer was provided in 2002 by Dr Croce’s laboratory [19]. In particular, the loss of two miRNA genes, miR-15a and miR-16-1, was found to be related to the onset of chronic lymphocytic leukemia (CLL). Then, many other human miRNA-coding genes were found located at fragile sites and altered regions in cancers, thus suggesting an important role of these small molecules in human cancer pathogenesis [20]. Depending on the function of their target, miRNAs are classified as oncogenes (onco-miRNAs) or tumor suppressors (ts-miRNAs) [21]. Onco-miRNAs target tumor suppressor molecules and, when upregulated, can promote tumor onset and growth. Moreover, some onco-miRNAs are known as metasta-miRNAs: they affect tumor progression at later stages, modulating malignant cell migration, invasion, and metastasization. Conversely, ts-miRNAs target oncogenes, and their downmodulation can determine malignant progression. Therefore, since miRNAs provide a strong contribution to all cancer-related pathways, they can be considered good candidates as clinical biomarkers and targets for therapies. Indeed, many platforms have been developed to evaluate genome-wide miRNA expression in different types of tumors, creating the possibility to identify signatures associated with diagnosis, progression, staging, response to treatment, and prognosis, thus opening a new wide research field concerning cancer therapy [21,22,23,24].

### 1.4. MiRNAs and Breast Cancer

The first miRNA expression profile of breast cancer was described by Iorio et al. in 2005 [15]. The analysis was performed on a cohort of 76 human breast cancer tissue samples compared to 10 healthy samples. The results showed 29 miRNAs significantly deregulated between normal and cancer tissue, and 15 were able to discriminate between normal and tumor tissues with 100% accuracy. Among the differentially expressed miRNAs, miR-10b, miR-125b, and miR-145 were downmodulated whilst miR-21 and miR-155 were upmodulated. Moreover, they identified miRNAs whose expression was correlated with specific breast cancer pathological features, such as estrogen and progesterone receptor status, nodal metastases, vascular invasion, proliferation index, and p53 immunohistochemical detection. An increasing number of studies published over the past 15 years have suggest that specific miRNAs could be clinically relevant as predictive and prognostic markers, potential therapeutic targets, or therapeutic tools in breast cancer.

Recent studies have proved that miRNAs could contribute to the crucial process of formulating a diagnosis. Indeed, it is well known that an early and accurate diagnosis is fundamental to choosing the most suitable therapeutic strategy and, consequently, to improving patient prognosis. Concerning breast cancer, for instance, the evaluation of the proliferation index Ki-67 by immunohistochemical assay is one of the analyses performed to distinguish between luminal A (Ki-67 score < 20%) and luminal B subtypes (Ki-67 score ≥ 20%) [25]. Although these criteria are used in the clinical practice, there are still some uncertainties due to the variability between different laboratories [26]. Søkilde and colleagues found a specific miRNA signature that could be a surrogate of Ki-67 in order to facilitate the diagnosis [27]. In particular, they demonstrated that luminal A tumors are characterized by a higher expression of miR-99a/let-7c/miR-125b-2 clusters in comparison with luminal B samples, thus proposing the evaluation of its expression level as a tool that could make the difference in clinical choices.

MiRNAs could be not only exploited as diagnostic biomarkers; they can also have a predictive and prognostic significance in breast cancer patients. For example, concerning luminal B HER2+ breast cancer, it has been demonstrated that miR-718, miR-4516, miR-210, and miR-125b-5p are specifically associated with chemo-sensitivity; miR-222 and let-7g correlate with pathological response; and high levels of miR-125b-5p during neoadjuvant chemotherapy treatment predicted a poorer disease free survival [28]. In addition, in a recent study, Pan JK et al. demonstrated that miR-211 drives early brain metastasis in triple-negative breast cancer mice models by modulating adherence to the blood–brain barrier and stemness properties [29]. In addition, the authors found high levels of miR-211 in plasma samples of in vivo models, suggesting its possible role as a circulating biomarker for the early detection of brain metastasis, thus indicating poor prognosis.

In the last decades, an increasing body of evidence has suggested that miRNAs could also be exploited as therapeutic targets [30]. Indeed, as previously mentioned, some overexpressed miRNAs play a tumor-promoting role in different cancer types. They could thus be targeted to induce an impairing effect on tumor growth and progression. In fact, in the last years, an increasing number of preclinical studies suggest that onco-miRNA inhibition, achievable with different techniques such as anti-miRNA oligonucleotides (AMOs) and miRNA sponges [31], could be a promising therapeutic strategy in breast cancer. Many studies describe the oncogenic value of miR-21, demonstrating that its suppression has a strong effect on tumor growth in breast cancer models. In addition, Bahreyni and colleagues have recently demonstrated that anti-miR-21 could be a valid adjuvant therapeutic tool in combination with chemotherapy for breast cancer treatment [32]. In particular, they delivered anti-miR-21 and epirubicin by a novel system, which consisted of a biocompatible and biodegradable tumor-targeted nanocomplex, in cancer in vivo models, observing a significant synergistic effect on tumor growth inhibition. Furthermore, the inhibition of miR-21 in breast cancer has also been investigated by Devulapally and coworkers [33]. In particular, they simultaneously inhibited the two different onco-miRNAs miR-21 and miR-10b, and observed that when using antisense-miR-21 and antisense miR-10b together, delivered by nanoparticles specifically directed against triple-negative breast cancer cells, tumor growth was significantly reduced both in in vitro and in vivo models. Similarly, Zhang and colleagues demonstrated that miR-155-inhibition, achieved by anti-miR-155 delivered in breast cancer cells by an innovative poly-antioxidant nanoplatform, has a strong anti-metastatic effect, in both in vitro and in vivo models, synergistically with the antioxidant action of the delivery system [34]. In addition, it was demonstrated that the concurrent inhibition of miR-21 and miR-155 expression was able to impair tumor cell proliferation and metastasis through a combinatorial effect with photodynamic therapy [35]. However, despite preclinical pieces of evidence, no miRNA-based therapy has yet received clinical approval to be tested on breast cancer patients in a phase III clinical trial [30].

Last but not least, many studies describe the possible role of tumor-suppressing miRNAs as therapeutic tools in breast cancer treatment. For example, Ma J. and colleagues observed that miR-302b expression levels are low in breast cancer, suggesting its role as ts-miRNA [36]. Moreover, our group demonstrated that miR-302b-overexpression enhanced sensitivity to the chemotherapeutic drug cisplatin by inhibiting the E2F family, in particular E2F1, a crucial transcription factor that regulates G1/S transition, by modulating the DNA damage response pathway in breast cancer models [37]. In addition, we investigated the possible role of miR-302b as new therapeutic tool in a combinatorial approach with cisplatin in in vivo models [38], demonstrating that the combination of miR-302b delivery and cisplatin treatment significantly impaired tumor growth through a complex axis that involved ITGA6, the E2F family, and YY1, in in vivo xenografted models of triple-negative breast cancer. Another ts-miRNA poorly expressed in breast cancer is miR-4731-5p [39]. It was found that miR-4731-5p overexpression inhibited glycolysis, migration, and invasion processes in both in vitro and in vivo breast cancer models by directly targeting the oncogene PAICS and thus reducing the PAICS-induced phosphorylation of FAK, a fundamental promoter of glycolysis and growth factors. Similarly, also miR-138 has a tumor-suppressive role; it was demonstrated to inhibit the proliferation of breast cancer cells by promoting apoptosis and modulating NF-κB/VEGF pathways [40]. These are only few examples that demonstrate how miRNAs could concretely change the way of managing breast cancer.

## 2. MiRNAs and HER2+ Breast Cancer

### 2.1. Functional Roles of MiRNAs in Tumor Formation, Progression, Metastasis, and Therapy Resistance 

#### 2.1.1. MiRNAs Regulated by HER2 and Dysregulated in HER2-Positive Breast Cancer

The role of miR-33b in HER2+ breast cancer primary tumors was investigated by Pattanayak and coworkers. This group demonstrated that, in patient-derived HER2+ tumor samples and HER2+ breast cancer cell lines, miR-33b was significantly downregulated. Upon transfection with miR-33b, EMT, proliferation, invasion and migration were found to be reduced, whereas apoptosis was upregulated in HER2+ breast cancer cell lines [41]. Since the enhancer of zeste homolog 2-gene (EZH2) was downregulated following miR-33b transfection, the authors investigated this connection. They found that EZH2 downmodulation was due to the miR-33b-mediated targeting of MYC, which was already known to bind the EZH2 promoter. Thus, a novel miR-33b/MYC/EZH2 axis was described [41].

Amorim et al. performed a next-generation sequencing assay to quantify miRNA and circRNA levels in the luminal cell line HB4a and C5.2, which is an HER2-overexpressing clone of HB4a, and their extracellular vesicles [42]. In contrast to stable levels of circRNA, the levels of 16 miRNAs were modified in the HER2-overexpressing clone. Using data from the TCGA, the authors demonstrated a possible involvement in the HER2-axis of miR-223-3p, miR-421, and miR-21-5p.

Studies performed by Gorbatenko et al. aimed to identify differences of the miRNA expression profile induced by HER2 and its truncated form p95HER2 in HER2-/p95HER2-overexpressing MCF-7 breast cancer cells [41]. They found distinct changes in the expression profile and demonstrated a shift towards an expression pattern typical of the basal breast cancer subtype upon p95HER2-overexpression. Among the significantly altered miRNAs, they found miR-221 and miR-222. It was shown that by reducing the expression of estrogen receptor α in p95HER2-overexpressing cells, these two miRNAs impaired the response to anti-hormonal therapy. Together with miR-503, which was also found upregulated in p95HER2-overexpressing MCF-7 cells, these three miRNAs were shown to downregulate Myb proto-oncogene-like 1 (MYBL1) protein expression. Then, upregulation of miR-221/222 and -503 could increase cell mobility. Taken together, these data showed that altered miRNA expression is co-responsible for the more aggressive p95HER2 tumor subtype.

The role of the circular RNA circCDYL in HER2-negative breast cancer was reported by Liang and colleagues. Overexpression of circCDYL led to proliferation and activation of autophagosome formation by sponging miR-1275 [43]. The levels of this circular RNA were also found to be elevated in tissue samples of HER2+ patients compared to adjacent normal tissues [44]. Interestingly, the ectopic modulation of circCDYL had no effect on autophagocytosis in the HER2+ breast cancer cell line SkBr3. Furthermore, circRNA pull-down experiments did not reveal an interaction with miR-1275 in SkBr-3 cells as was seen in HER2-negative cell lines. In any case, the overexpression of circCDYL promoted the activation of the PI3K/Akt pathway in SkBr3 cells. To identify other circCDYL-binding miRNAs, a miRNA deep-sequencing was performed, which revealed an interaction with miR-92b-3p, also confirmed by miRNA pull-down. This interaction most likely was not due to the classical miRNA sponging, since miR-92b-3p target genes are not affected by circCDYL silencing. Vice versa, a dual luciferase reporter assay confirmed that miR-92b-3p targets circCDYL. Moreover, miR-92b-3p overexpression reduced cell proliferation in vitro and was shown to be associated with clinical outcome of HER2+ breast cancer patients.

#### 2.1.2. MiRNAs Involved in Anti-HER2 Therapy Resistance

Luo et al. revealed that, in HER2+ breast cancer cell models sensitive to trastuzumab, IGF2/IGF-1R/IRS1 oncogenic signaling is kept at bay by a negative feedback loop that involves the two IRS1-targeting miRNAs, miR-128-3p and miR-30a, and an IGF2-targeting miRNA, miR-193-5p. The expression of these three miRNAs is induced by FOXO3a, which is activated by levels of IGF2. In resistant cells, this negative feedback loop is disrupted and IGF2 and IRS1 are upregulated [45].

The intronic miRNA-4728 originates from an excised intron of the HER2 pre-mRNA, and it has been demonstrated that both its mature forms (miR-4728-5p and miR-4728-3p) are functional and significantly upregulated in HER2+ breast cancer patients [46,47]. Indeed, a luciferase reporter assay revealed that miRNA-4728-5p directly targets ErbB3-binding protein 1 (EBP1), which suppresses HER2 by binding to its promoter region [46,48]. Through this targeting, miR-4728-5p triggers a positive feedback loop to promote HER2-dependent tumor progression. In addition, recent studies proved that miR-4728-3p downregulates estrogen receptor 1 (ESR1) [47,49]. The reduced ESR1 expression prevents ESR1-mediated transcription of NOXA, an endogenous inhibitor of the antiapoptotic MCL-1 [50]. In conclusion, although miR-4728 is able to promote cell survival and to reduce the sensitivity to hormonal therapy by ESR1 downmodulation, it increases response to anti-HER2 therapy through the direct targeting of EBP1. Another intronic miRNA within the *ERBB2* gene was discovered by Shabaninejad et al. and named HER2-miR1 [51]. The presence of this novel miRNA was confirmed in several human cell lines. Upon ectopic overexpression of this miRNA, the Wnt signaling pathway was downmodulated, indicating an oncosuppressive role.

Inspired by one of the first ongoing phase I clinical trials using targo-miRs (minicells loaded with miR-mimics) as therapeutic tools in human subjects with malignant pleural mesothelioma (NCT02369198 [52]), Normann et al. investigated the role of miRNAs in combination with HER2-targeting drugs in vitro [53]. In their study, eight miRNAs were identified that sensitized KPL4 and SUM190PT cell lines to treatment with trastuzumab or lapatinib. In particular, higher expression levels of miR-101-5p were associated with breast cancer-specific survival (BCSS) and overall survival (OS). Upon these findings, the authors suggested to investigate the combination of targeted drugs with miRNAs to improve the current treatment options in HER2+ breast cancer. Indeed, an additional article of the same group focused exclusively on the role of miR-101-5p in HER2+ breast cancer [54]. Data from the TCGA and METABRIC dataset confirmed the downregulation of miR-101-5p in HER2+ patients and in vitro experiments using KPL4 cells showed an enhanced response to trastuzumab upon ectopic overexpression of miR-101-5p. The transfection of miR-101-5p combined with anti-HER2 treatment led to several changes in the proteomic landscape of KPL4 cells, including PI3K-Akt, mTOR, and ErbB signaling.

Exosome-delivered miR-567 was shown to reverse trastuzumab resistance by Han and colleagues [55]. MiR-567-containing exosomes isolated from MCF-10 cells were incubated with trastuzumab-resistant SkBr-3 and BT-474 (SkBr-3R and BT-474R) cells. Intracellular uptake of the exosomes was proven via immunofluorescence and the subsequent improved response to trastuzumab was shown by western blot and cell viability assay. Furthermore, autophagy-related protein 5 (ATG5) was identified as a direct target of miR-567 by luciferase reporter assay. Since ATG5 is well known to be associated with carcinogenesis, the authors suggested that the reversed trastuzumab resistance might be due to reduced protein expression of ATG5 upon miR-567 overexpression.

Other dysregulated miRNAs in trastuzumab resistant cell lines were identified by Rezaei and coworkers [9]. Seven candidate miRNAs were evaluated in sensitive and trastuzumab-resistant BT-474 cells, and miR-23b-3p, miR-195-5p, miR-656-5p, and miR-340-5p were found to be significantly dysregulated in the resistant cell lines. Using TargetScan and miRDB, several putative targets associated with drug resistance pathways were identified.

Taking all these data together, miRNAs have shown a variety of functions in HER2+ breast cancer (Figure 1, created with BioRender.com, accessed on 21 September 2022). Their overexpression can support or suppress oncogenic pathways, a disrupted negative feedback loop may lead to cancer progression, or their aberrant expression can have an impact on the response to targeted therapies. Many studies also describe an improvement in therapy response upon ectopic miRNA modulation, indicating a therapeutic use when co-administered with HER2-targeted therapeutics (Table 1).

### 2.2. MiRNAs as Biomarkers in HER2+ Breast Cancer

#### 2.2.1. MiRNA Signatures for HER2+ Breast Cancer Diagnosis

Since breast cancer can be classified through a gene expression analysis, it is not unexpected that miRNAs also show a specifically altered expression profile in different breast cancer subtypes [57]. In fact, concerning HER2+ breast cancer, high levels of circulating miR-375 and low levels of circulating miR-122 were associated with HER2-status [58]. In addition, Souza et al. identified cell-free circulating miRNAs for early detection of breast cancer, and also distinguished between different subtypes [59]. In particular, they found that four circulating miRNAs—miR-548ar-5p, miR-584-3p, miR-615-3p, and miR-1283—had a significantly differential expression in the HER2-enriched subtype.

#### 2.2.2. MiRNAs as Prognostic and Predictive Biomarkers in HER2+ Breast Cancer

An increasing body of evidence underlines the potential of several miRNAs to predict therapy response in HER2+ breast cancer. Since approximately 50% of the patients are refractory to the anti-HER2-based standard treatment, there is an important clinical need to discriminate between responsive and non-responsive patients in order to choose the best clinical option, especially in the metastatic setting. In this regard, it has been demonstrated in different cancer types that miRNA expression profiles differ depending on the capability to respond to therapy, thus representing promising predictive tools. In particular, miRNAs available in easily accessible specimens, such as liquid biopsies, hold promising potential as non-invasive biomarkers. In HER2+ breast cancer, Li and colleagues identified a four miRNA signature as a serum biomarker that may predict the therapeutic efficacy of the trastuzumab regimen in patients with HER2+ metastatic breast cancer [60]. In particular, they demonstrated that the overexpression of miR-451a, miR-16-5p, and miR-17-3p, as well as the downmodulation of miR-940, are significantly associated with prognosis and therapy response. Moreover, they found that miR-940, miR-451a/miR-16-5p, and miR-17-3p target PTEN, IGF1R, and SRC mRNA, respectively, whose aberrant signaling was described as a mechanism involved in trastuzumab resistance of breast cancer cells [58,59,60,61,62].

A prospective collection of plasma and tissue samples at baseline (T0), after two weeks of trastuzumab (T1), and at surgery (T2) was included in the study protocol of the NeoALTTO trial (NCT00553358). The aim of this open-label, multicenter phase III study was to evaluate dual HER2 inhibition by the combination of lapatinib and trastuzumab in a neoadjuvant setting [63]. The primary endpoint of this study was pCR and the secondary endpoint was EFS. A group of our institute led by Di Cosimo obtained access to these samples and performed a profiling of 752 miRNAs in plasma samples at T0 and T1 [64]. They identified four miRNA signatures with a predictive value for the achievement of pCR. None of these signatures was found to be associated with EFS. Nevertheless, when performing a univariate analysis, circulating tumor (ct)-miR-140-5p, belonging to the signature at T1 of the trastuzumab arm, was significantly associated with EFS (*n* = 127; HR = 0.73; 95% CI, 0.59–0.91). The continuation of the analysis of the dataset obtained from the plasma samples deriving from the NeoALTTO trial was reported one year later, considering only the 52 patients enrolled in the trastuzumab arm and focusing on early changes of circulating miRNA levels from T0 to T1 [65]. Changes in ct-miR-148a-3p and ct-miR-374a-5p levels were significantly associated with pCR (Kw test *p*-values of 0.008 and 0.048, respectively). Considering jointly “high changes” of ct-miR-148a-3p from T0 to T1 and high ct-miR-140-5p levels at T1, which had been previously shown to be associated with both pCR and EFS, the chance of achieving pCR was further enhanced up to 54%. Using GO and KEGG analysis, ct-miR-140-5p, ct-miR-148a-3p, and ct-miR-374a-5p were revealed to have several target pathways in common, including cell metabolism regulation, AMPK and MAPK signaling, and progression of hepatocellular carcinoma, indicating that they could also hold a functional role. Regarding the tissue samples at T0 in the trastuzumab arm, our group found two different two-miRNA signatures with a prognostic and predictive value [66]. The prognostic signature comprised miR-153-3p and miR-219a-5p, and showed a significant association with EFS. Another signature consisting of miR-215-5p and miR-30c-2-3p was shown to interact with pCR for EFS. The predictive signature included miR-31-3p and miR-382-3p, and was significantly associated with pCR.

The role of miRNAs in plasma-derived exosomes was investigated by Stevic et al. in TNBC and HER2+ breast cancer patients [67]. A panel of 45 miRNAs was measured in exosomes derived from a cohort of 435 breast cancer patients enrolled in the GeparSixto trial (NCT01426880, 211 HER2+ and 224 TNBC), revealing significant differences in the expression pattern according to their different tumor biology. Exosomal miR-27b was found to be significantly higher in HER2+ patients compared to TNBC patients, and was shown to predict pCR. On the other hand, miR-422a was found to be downmodulated in HER2+ breast cancer and upmodulated in TNBC exosomes. Furthermore, exosomal miR-16, miR-328, and miR-660 were shown to be associated with lymph node status in HER2+ patients.

The group of Liu et al. assessed the expression of several miRNAs selected by literature research via quantitative real-time PCR in serum samples of 83 early HER2+ breast cancer patients receiving a neoadjuvant chemotherapy combined with trastuzumab [68]. Serum samples were collected before therapy start (BL), at the end of the second cycle (FDN), and at surgery (SDN). Out of nine serum-miRNAs included in the study, circulating miR-21 (called ser-miRNA-21) was significantly reduced in the responding patients, both at FDN and SDN. Interestingly, changes of ser-miR-21 (from BL to FDN or BL to SDN) were also shown to be significantly associated with overall survival and disease-free survival, whereas there was no association with the expression level at the single timepoints. Given these data, ser-miR-21 could be a potential biomarker to predict clinical response to neoadjuvant chemotherapy combined with trastuzumab, but it may also serve as a prognostic marker in HER2+ breast cancer patients.

A study including 65 HER2+ breast cancer patients receiving neoadjuvant chemotherapy combined with trastuzumab was performed by Zhang’s group [69]. The baseline level of serum-miR-222-3p was assessed via quantitative real-time PCR; it was found that higher levels of serum-miR-222-3p were associated with an inferior pCR rate. Regarding the prognostic value of this miRNA, patients with a low serum-miR-222-3p expression had a superior DFS and OS. MiR-222-3p has been previously shown to be involved in several cardiovascular processes [70]. In this study, it was also shown to act as a potential serum biomarker for side effects such as cardiotoxicity and anemia upon trastuzumab-containing neoadjuvant therapy.

The oncosuppressive miR-205 has been previously described as targeting the HER3 receptor, thus increasing the response to the tyrosine kinase inhibitors lapatinib and gefitinib in preclinical models [71]. In a further study it was demonstrated that the suppression of HER3 by ectopic overexpression of miR-205 increased responsiveness to trastuzumab in vitro in HER2+ cell lines [72]. These data have also been confirmed in a patient-derived-xenograft (PDX) model. Additionally in a set of 52 HER2+ breast cancer patients receiving adjuvant, trastuzumab-based therapy, higher levels of miR-205 were significantly associated with increased disease-free-survival.

Interesting insights on the miRNA mechanism of action and regulation came from the study of miRNA and lncRNA crosstalk. For example, Müller and colleagues investigated two known interacting lncRNA/miRNA couples, specifically H19/miR-675 and NEAT/miR-204, in plasma samples of 63 breast cancer patients and 10 healthy donors [73]. The relations between these couples of ncRNAs were confirmed in this context. This crosstalk are particularly relevant in HER2+ breast cancer samples, compared with both healthy donors and with the other breast cancer subtypes. Finally, functional studies in in vitro models proved that H19/miR-675 and NEAT/miR-204 impact cell proliferation and apoptosis. Table 2 summarizes the literature described in this chapter.

## 3. Conclusions

Overall, we can state that, in a scenario where the response rate to trastuzumab-based drugs is still of approximately 50% and given the severe side effects of most anti-HER2 therapies, miRNAs have promise to be versatile clinical tools, potentially serving both as therapeutic options and reliable biomarkers

Tissue or ct-miRNAs might help the treatment decision process by identifying patients more likely to be responsive to a specific compound; this would allow avoiding waste of time and unnecessary side effects for resistant cases, for whom alternative or combination strategies should be proposed, as well as to potentially de-escalate the therapy regimen for responsive patients.

Even though the validation of miRNA signatures is often hampered by the use of different platforms and non-standardized procedures, the data available to date strongly suggest that their application in the clinical practice is a concrete possibility.

On the contrary, it seems that the application of a miRNA-based therapy has more challenging issues to face, such as delivery efficiency and specificity, and limitation of side effects. However, even though no clinical trials are currently ongoing in breast cancer, several groups are conducting successful preclinical studies to define the best delivery strategies.

In conclusion, current and future studies will generate more evidence in order to develop clinically approved miRNA-based biomarkers and novel therapeutic agents in HER2+ breast cancer.

## Figures and Tables

**Figure 1 cancers-14-05326-f001:**
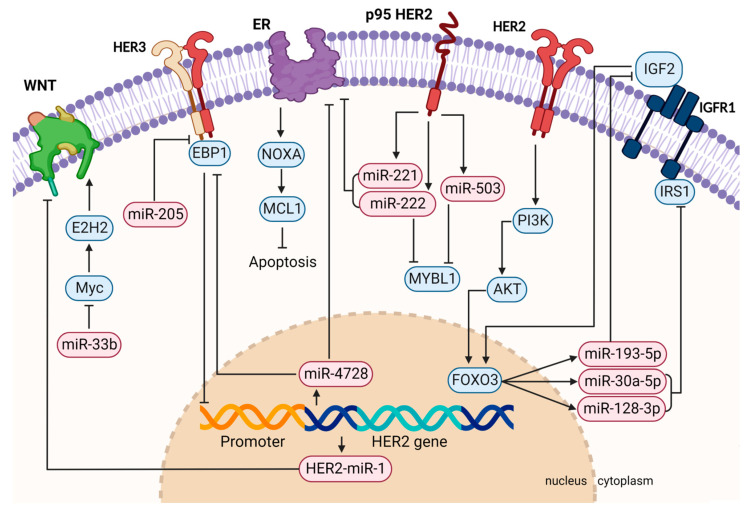
Graphical summary of HER2-related pathway regulation by miRNAs involved in tumor formation, progression, metastasis, and therapy resistance. Aberrantly activated signaling pathways can lead to differently expressed miRNAs, causing the deregulation of their target mRNAs. For instance, miR-221/222 are activated downstream of HER2 and are able to directly target ER. HER2-activated FOXO3 also regulates the IGFR1 pathway by inducing the expression of miR-193-5p, miR-30-5p, and miR-128-3p. Vice versa, miRNAs can also regulate members of the HER family themselves, such as HER3, which is directly targeted by miR-205. This figure was created with BioRender.com.

**Table 1 cancers-14-05326-t001:** Functional role of miRNAs in HER2+ breast cancer.

miRNA	Reference	Target	Functional Role
**miR-128-3p, miR-30a-5p,** **miR-193-5p**	Luo et al. (2021) [45]	IRS1 and IGF2	Regulation of IGFR1 pathway
**miR-33b**	Pattanayak et al. (2020) [56]	MYC	Promotion of apoptosis and reduction of invasion and migration
**HER2-miR1**	Shabaninejad et al. (2022) [51]	Wnt pathway	Oncosuppressive intronic miRNA
**miR-4728**	Zhou et al. (2021), Rui et al. (2022), Lu et al. (2011), Floros et al. (2018) [46,47,48,49]	EBP1 and ESR1	Promotion of cell survival and response to anti-HER2 therapy; reduction of sensitivity to hormonal therapy
**miR-221,** **miR-222, miR-503**	Gorbatenko et al. (2019) [41]	ESR1 and MYBL1	Reduction of response to endocrine therapy and increase of cell mobility
**miR-101-5p, miR-518a-5p, miR-19b-2-5p, miR-1237-3p, miR-29a-3p, miR-29c-3p, miR-106a-5p, miR-744-3p**	Normann et al. (2021) [53]		Promotion of response to anti-HER2 therapy and prognostic role
**miR-101-5p**	Normann et al. (2022) [54]		Downregulated in HER2+ breast cancer patients
**miR-567**	Han et al. (2020) [55]	ATG5	Promotion of response to anti-HER2 therapy
**miR-23b-3p, miR-195-5p, miR-656-5p,** **miR-340-5p**	Rezaei et al. (2019) [9]	Drug resistance pathways	Dysregulated in resistant cells
**miR-92b-3p**	Liang et al. (2021) [44]	circCDYL	Reduction of cell proliferation and prognostic role

**Table 2 cancers-14-05326-t002:** miRNAs as biomarkers in HER2+ breast cancer.

miRNA	Reference	Value in HER2+ Breast Cancer
**miR-451a, miR-16-5p, miR-17-3p, miR-940**	Li H et al. (2018) [60]	Predictor of response and prognostic role
**miR-27b, miR-433, miR-16,** **miR-328, miR-660, miR-422a**	Stevic et al. (2018) [67]	Predictive role, association with N-status, distinction between TNBC and HER2+ BC
**miR-21**	Liu et al. (2019) [68]	Predictor of response
**miR-222-3p**	Zhang et al. (2020) [69]	Predictive and prognostic role and association with cardiotoxicity and anemia, side effects of trastuzumab therapy
**miR-205**	Cataldo et al. (2018) [72]	Predictor of response and prognostic role
**miR-675, miR-204**	Müller et al. (2019) [73]	Indicators of disease aggressiveness
**miR-100-5p, miR-374a-5p, miR-574-3p, miR-140-5p, miR-328-3p, miR-145-5p, miR-34a-5p, miR-98-5p, miR-100-5p,** **miR-144-3p, miR-362-3p,** **miR-197-3p, miR-320c, miR-100-5p,** **miR-376c-3p, miR-874-3p,**	Di Cosimo et al. (2019) [64]	Predictive and prognostic role
**miR-148a-3p, miR-374a-5p**	Di Cosimo et al. (2020) [65]	Predictive role
**miR-215-5p,** **miR-30c-2-3p,** **miR-153-3p,** **miR-219a-5p,** **miR-31-3p, miR-382-3p**	Pizzamiglio and Cosentino et al. (2021) [66]	Predictive and prognostic role

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
