# Peer review of "The Role of MicroRNAs in HER2-Positive Breast Cancer: Where We Are and Future Prospective"

_cancers, 2022, doi:10.3390/cancers14215326_

Round 1

Reviewer 1 Report

In this manuscript, Fogazzi et al summarized current status and problems in clinical treatment and outcomes of HER2-positive breast cancer and reviewed recent research progress in understanding microRNAs in regulating cancer biology, therapy response, diagnosis, and prognosis of HER2-positive breast cancer. Overall, this is a comprehensive review of literature related to microRNA research in breast cancer with an emphasis on HER2-positive breast cancer. A number of review articles related to microRNA in breast cancer have previously been published. However, this is the first review article about the role of microRNA in HER2-positive breast cancer. Given that anti-ErbB2 therapy resistance is a problem and microRNAs may serve as predictive/diagnostic markers and therapeutic targets for this subtype of cancer, this review may be of interest to readers in this field.

The manuscript is well-written and the review of the literature is comprehensive. I only have several minor suggestions.

1.     In Chapter 2.1. it will be better to separate the paragraphs into two sections with first one to discuss microRNAs that are regulated by HER2 and dysregulated in HER2-positive BC, and second one to discuss microRNAs that are involved in anti-HER2 therapy resistance.

For miRNA-4728 on page 6, please cite Newie I et al. paper that first described HER2 encoding miT-4728-3p. (PLoS ONE 9(5): e97200)

For Figure 1, please provide brief description about how the signaling pathways regulate miRNAs and are regulated by miRNAs.

2.     For table 1. It would be better to divide the table into two tables. First table could describe microRNAs in regulating HER2-positive BC tumor formation, progression, metastasis, and therapy resistance. The first table should be inserted in Chapter 2.1. Second table could describe microRNAs as biomarkers and be inserted in Chapter 2.2. In addition, the readers could be benefitted from description of targets, pathways, and roles (particularly for HER2-positive breast cancer biology, therapy resistance, diagnosis and prognosis) for each of the microRNAs in the tables. The current description in the table contains citations of what is already described in the text and is not easy to grasp what these microRNAs and their functions are and how they can be used for diagnosis and therapy purposes.

3.     The title is about where we are and future prospective. However, there is no discussion about future prospective. Please conjure one.

Reviewer 2 Report

The review article by Fogazzi et al. summarizes the recent literature regarding the role of miRNAs in HER2+ breast cancer. Micro RNAs are a class of post transcriptional gene regulators with critical functions in normal cellular processes as well as disease processed. They are usually 18-23 nucleotides in length. HER2, a tyrosine kinase receptor is overexpressed in about 15-20% of all breast cancer patients and usually associated with poor prognosis.  Despite improved clinical outcomes with several FDA approved anti-HER2 drugs, acquired drug resistance occurs in many patients triggering recurrence. Hence, identification of new predictive biomarkers for better treatment are needed. Micro RNA may have a regulatory role in multiple cells proliferation, and cell cycle progression pathways of breast cancers as shown by various studies. Hence, miRNA can be considered good candidates as clinical biomarkers and targets for therapies. The article is well written and has included all the latest literatures and the various updates regarding the micro-RNA’s in HER2+ BC.  The article captures the latest literatures, and the flow of the information is well written and easy to follow. The article is divided into sub section which makes the flow of information easy to understand.

It would have been better if the authors could have made a table with the ongoing trials (if any) with the micro RNA’s in HER2+ BC? Overall,  a well written article.

Reviewer 3 Report

Well written review of role of miRNAs in Her2 positive cancer and authors  covered all potential prospect of miRNAs in management of breast cancer both as biomarkers and therapeutic targets. 

Author Response

We thank the reviewer for the positive evaluation.